# Synthesis, Structural Analysis, and Peroxidase-Mimicking Activity of AuPt Branched Nanoparticles

**DOI:** 10.3390/nano14131166

**Published:** 2024-07-08

**Authors:** Silvia Nuti, Javier Fernández-Lodeiro, Jose M. Palomo, José-Luis Capelo-Martinez, Carlos Lodeiro, Adrián Fernández-Lodeiro

**Affiliations:** 1BIOSCOPE Research Group, LAQV-REQUIMTE, Chemistry Department, NOVA School of Science and Technology (FCT NOVA), Universidade NOVA de Lisboa, 2829-516 Caparica, Portugal; s.nuti@campus.fct.unl.pt (S.N.); jlcm@fct.unl.pt (J.-L.C.-M.); cle@fct.unl.pt (C.L.); 2PROTEOMASS Scientific Society, Praceta Jerónimo Dias, Num. 12, 2A, Sto António de Caparica, 2825-466 Costa de Caparica, Portugal; 3Instituto de Catalisis y Petroleoquimica (ICP), Consejo Superior de Investigaciones Científicas (CSIC) Marie Curie 2, 28049 Madrid, Spain; josempalomo@icp.csic.es (J.M.P.)

**Keywords:** platinum, gold, bimetallic, nanoparticles, Au@Pt, nanozyme, catalysis, TMB

## Abstract

Bimetallic nanomaterials have generated significant interest across diverse scientific disciplines, due to their unique and tunable properties arising from the synergistic combination of two distinct metallic elements. This study presents a novel approach for synthesizing branched gold–platinum nanoparticles by utilizing poly(allylamine hydrochloride) (PAH)-stabilized branched gold nanoparticles, with a localized surface plasmon resonance (LSPR) response of around 1000 nm, as a template for platinum deposition. This approach allows precise control over nanoparticle size, the LSPR band, and the branching degree at an ambient temperature, without the need for high temperatures or organic solvents. The resulting AuPt branched nanoparticles not only demonstrate optical activity but also enhanced catalytic properties. To evaluate their catalytic potential, we compared the enzymatic capabilities of gold and gold–platinum nanoparticles by examining their peroxidase-like activity in the oxidation of 3,3′,5,5′-tetramethylbenzidine (TMB). Our findings revealed that the incorporation of platinum onto the gold surface substantially enhanced the catalytic efficiency, highlighting the potential of these bimetallic nanoparticles in catalytic applications.

## 1. Introduction

Catalysis plays a crucial role in numerous industrial and chemical processes, enabling reactions to occur more efficiently and often under milder conditions than would otherwise be possible [1].

In recent years, nanomaterials have emerged as highly effective nanocatalysts, offering unique properties that enhance catalytic performance. These nanocatalysts often exhibit higher surface area-to-volume ratios, increased active sites, and improved stability, compared to their bulk counterparts. Metals such as gold (Au), platinum (Pt), or palladium (Pd) are commonly used in the synthesis of nanocatalysts, due to their excellent catalytic properties [2]. One of the well-known applications of these nanomaterials is their ability to mimic the activity of natural enzymes, a property that has gained significant interest recently. These nanozymes exhibit enzyme-like catalytic behavior, offering several advantages over their natural counterparts. Unlike natural enzymes, which can be costly and exhibit limited stability under varying environmental conditions, nanozymes are typically more robust and can be engineered to perform specific catalytic functions with high efficiency and stability [3].

It has been observed that the inner layers of structured nanomaterials may remain inaccessible for catalytic reactions. Given the scarcity of platinum and its widespread use in numerous chemical processes, optimizing its utilization to achieve maximum efficiency with minimal usage is essential [4]. In this sense, bimetallic nanoparticles (NPs) have garnered significant attention in recent years, owing to their unique ability to merge the distinct properties of two different metal compositions within a single nanoparticle. Adding a second metal can produce novel synergies, enhancing performance across various domains, not only in the catalysis field [5] but also in biomedicine [6], magnetism [7], optics [8], and beyond. Gold nanoparticles (Au NPs), specifically anisotropic NPs, have demonstrated their efficiency in different fields over the years, due to their unique characteristics, including tunable optical properties, robustness, and stability to oxidation [9]. Among them, Au branched NPs, such as Au nanostars, show an intense and tunable plasmon band in the Vis-NIR region (in between 560 nm and 1260 nm, approximately) [10,11], and thanks to their structure, they produce intense scattered electric fields located at the tips [12]. Plasmonically active nanomaterials based on gold–platinum nanoparticles (AuPt NPs) have attracted attention due to the possibility of combining the exceptional catalytic properties of Pt with the characteristics of Au, namely its plasmonic properties and the variety of shapes and sizes obtainable at the nanoscale. Pt possesses, in fact, weak plasmonic properties, which makes the combination with more efficient plasmonic materials like Au a good alternative to obtain plasmonically active nanoparticles. Such nanomaterials have been employed recently in a variety of applications, including photocatalysis [13], sensing [14], drug delivery [15], and chemotherapy [16]. In this context, designing nanomaterials with a gold core and an outer layer of platinum can significantly enhance the effective use of platinum. This configuration not only maximizes the catalytic activity of the exposed platinum but also takes advantage of the unique properties of the gold nanoparticle.

The strategic combination of well-defined plasmonic properties, such as those exhibited by branched gold nanostars, with the incorporation of platinum on their surface, presents a promising solution for addressing different challenges [17]. The branched gold nanostars provide intense and tunable plasmonic responses. When platinum is added to the surface, it enhances the catalytic capabilities of the nanostars, leveraging the exceptional catalytic properties of platinum. This synergy not only maximizes the use of scarce platinum by concentrating it where it is most effective but also expands the functional utility of the nanomaterials. While the synthesis of anisotropic nanomaterials based on AuPt is more commonly documented, often employing Au nanorods as templates [18,19,20], there are also a few reports on branched AuPt bimetallic nanostructures [17,21,22,23] that exhibit branched or star-like morphologies. However, they are characterized by limited tunability, high damping in UV-Vis response, or reliance on different surfactants or organic solvents and elevated temperatures for their synthesis.

In this study, we present a straightforward approach, employing poly(allylamine hydrochloride) (PAH) to fabricate branched AuPt NPs in an aqueous setting at an ambient temperature. This method offers precise control over the NP size, localized surface plasmon resonance (LSPR) band, and branching degree, facilitated by a straightforward post-synthesis surface functionalization step. The extensive tunability of the LSPR band within the visible to near-infrared (Vis-NIR) spectrum, along with the post-functionalization of the NP surfaces with a negatively charged polymer, underscores the potential utility of this methodology for future applications. Finally, as a proof of concept of the improved catalytic performance of these bimetallic nanostructures, the potential of the synthesized AuPt branched NPs to act as peroxidase-mimicking nanozymes was investigated. Nanozymes, characterized by their enzyme-like properties, have emerged as a solution to address the inherent limitations of natural enzymes, including their high cost and low stability. Specifically, the ability of the AuPt branched NPs to catalyze the oxidation reaction of colorless 3,3,5,5-tetramethylbenzidine (TMB) to its blue oxidized form (ox-TMB) in the presence of hydrogen peroxide (H_2_O_2_) was evaluated. This reaction serves as a model to assess the nanozyme activity, providing insights into their potential applications in catalysis [24].

## 2. Experimental Section

### 2.1. Reagents

Gold(III) chloride trihydrate (HAuCl_4_·3H_2_O, 99.9%), potassium tetrachloroplatinate (K_2_PtCl_4_,98%), trisodium citrate dihydrate (C_6_H_5_Na_3_O_7_·2H_2_O, ≥99.5%), sodium borohydride (NaBH_4_, ReagentPlus^®^, Sigma-Aldrich, Saint Louis, MO, USA 99%), L-ascorbic acid (C_6_H_8_O_6_, BioXtra ≥ 99.0% crystalline), poly(allylamine hydrochloride) ([CH_2_CH(CH_2_NH_2_ · HCl)]_n_, average Mw 17,500 (GPC vs. PEG std.)), sodium hydroxide (NaOH, BioXtra ≥ 98%, pellets, anhydrous), hydrochloric acid (HCl, ACS reagent, 37%), sodium acetate anhydrous (C_2_H_3_NaO_2_ ≥ 99.5%), acetic acid glacial (CH_3_CO_2_H, ReagentPlus^®^, ≥99%), poly(acrylic acid-co-maleic acid) solution (PAcMA, average Mw 3000, 50 wt. % in H_2_O), and phosphate buffered saline (PBS) tablets were obtained from Sigma-Aldrich, Saint Louis, MO, USA. TMB (3,3′,5,5′-tetramethylbenzidine, 98%) was obtained from Thermo Scientific, Waltham, MA, USA. Hydrogen peroxide (H_2_O_2_, 30% *w*/*v*, 100 vol.) was obtained from Panreac AppliChem, Darmstadt, Germany. Dimethyl sulfoxide (DMSO, reagent grade ≥ 99.5%) was obtained from Honeywell. All reagents were used as received, without further purification. Ultrapure water (type I) was used for the preparation of all the water-based solutions. The glassware was cleaned with aqua regia and rinsed with ultrapure water prior to the experiments.

### 2.2. Synthesis of NPs

The synthesis of Au seeds of 3–5 nm stabilized with sodium citrate was based on a previous report [10]. Briefly, 20 mL of an aqueous solution containing 0.125 mM HAuCl_4_ and 0.25 mM trisodium citrate was prepared in a round-bottomed flask at RT. Under vigorous stirring, 300 µL of an ice-cold freshly prepared 0.01 M NaBH_4_ solution was rapidly injected. Vigorous stirring was maintained for 15 more seconds and then slowed down for 15 min at RT. The seed solution was heated for 60 min at 40 °C with slow magnetic stirring and then cooled with continued stirring to ensure the removal of excess NaBH_4_ before use. The seed solution was diluted to a final concentration of 0.1 mM in Au metal for the NP growth experiments.

For the synthesis of the branched bimetallic NPs, in a round-bottom flask containing water (ca. 17 mL), 4 mL of PAH 17.5 K (1 mg/mL) and 0.3 mL of HAuCl_4_ (20 mM) were added under vigorous magnetic stirring. Next, the pH was adjusted to ~5.7 with a NaOH (0.02 M) solution. After 5 min of vigorous stirring, 0.5 mL of ascorbic acid (20 mM) and, subsequently, the Au seeds were added. After 2 h, K_2_PtCl_4_ (20 mM) and AA (20 mM) were quickly added one after the other. The reactions were left under vigorous magnetic stirring overnight at room temperature (~22 °C). Finally, the NPs were purified by centrifugation at for 20–30 min and redispersed in water.

### 2.3. Functionalization of NPs with PAcMA

The functionalization of Au and AuPt NPs with poly(acrylic acid-co-maleic acid) was achieved following a previous report, with minor modifications [10]. PAcMA was activated with EDC/NHS before adding it to the AuNPs. Briefly, to 5 mL of an aqueous solution containing 12 µL (50 wt. % in H_2_O) of PAcMA, 5 mL of 60 mM EDC and 5 mL of 60 mM NHS were added. The reaction was maintained under ultrasound for 15 min and then under vigorous magnetic stirring for 2 h. The activated polymer solution was dripped onto the NP reaction, which was vigorously stirred for 4 h. Finally, the NPs were washed with PBS (pH = 7.4) and water.

### 2.4. Enzymatic Activity Studies

To calculate the steady-state enzymatic kinetic parameters of the Au branched NPs, various concentrations of TMB (from 5 to 90 mM in DMSO) were added to a buffer solution, always using 10 μg of nanozyme in each experiment (100 μL, 0.5 mM Au (0) of Au NPs). To measure the AuPt branched NPs enzyme activity, 5 μg of the enzyme was used. Into a 1 cm cuvette were added 2094 μL of acetate buffer solution (0.1 M, pH = 4.0), 100 μL of branched NPs (0.5 mM of Au (0)), and 50 μL of TMB. After the homogenization of the sample, 256 μL of H_2_O_2_ (30%) was quickly added. The mixture was homogenized again, and UV-Vis spectra at 652 nm every 10 s were recorded for 400 s. In the case of the study of H_2_O_2_, 50 μL of 20 mM TMB was employed, and various concentrations of H_2_O_2_ (between 0.25 and 4 M) were added. All the experiments were performed at room temperature.

### 2.5. Characterization

The extinction spectra were recorded using JASCO 770 UV-vis-NIR, JASCO 630 UV-vis, and a JASCO 650 UV-vis (Jasco Corporation, Tokio, Japan), and dynamic light scattering (DLS) and ζ-potential analysis were carried out in a Malvern ZS instrument provided by the PROTEOMASS-BIOSCOPE facility (Caparica, Portugal).

STEM and EDX analyses were obtained with a FEI Titan ChemiSTEM High Tension, 200 kV. For HRTEM imaging, samples were imaged with a JEOL JEM-2100 electron microscope (Hillsboro, OR, USA) operating a LaB6 electron gun at 200 kVs, and images were acquired with a “OneView” 4 k × 4 k CCD camera at minimal under-focus. Magnifications were 100 kX and 1MX. HRTEM sample support: CF400-Cu-UL carbon square mesh, Cu, 400 Mesh, UL.

XPS (X-ray photoelectron spectroscopy) was performed in an ESCALAB250Xi (Thermo Fisher Scientific, Waltham, MA, USA). Analysis area (field of view on the sample) was defined by the X-ray spot size. X-ray sources included monochromated tAlKα (h = 1486.68 eV) radiation, operated at 220 W and 14.6 kV and with a spot size of 650 μm. The XPS spectra were collected at pass energies of 100 eV and 40 eV for survey spectra and individual elements, respectively. The energy step for individual elements was 0.1 eV. The XPS spectra were peak-fitted using the Avantage Data Processing Software (Thermo Fisher Scientific, Waltham, MA, USA). For peak fitting, the Shirley-type background was used. All the XPS peaks were referenced to adventitious carbon C1s, with the C-C peak at 284.8 eV. Charge neutralization by means of electron and ion guns run at 120 and 70 μA emission currents, respectively, were applied. Samples were drop-casted on the gold-coated Si wafers and mounted on the sample holder using adhesive double-sided carbon tape.

Inductively coupled plasma–optical emission spectrometry (ICP-OES) studies were carried out in an ICPE-9000 multitype ICP emission spectrometer from Shimadzu (Kyoto, Japan) equipped with a nebulizing system and using optical emission spectroscopy for detection.

## 3. Results and Discussion

### 3.1. Synthesis and Characterization of Nanoparticles

We previously presented the versatile utilization of branched Au NPs, synthesized with poly(allylamine hydrochloride) (PAH), as a template for the growth of palladium (Pd), yielding branched AuPd NPs with customizable optical properties. These NPs exhibited enhanced catalytic activity, compared to their pure Au counterparts [25]. Recognizing the strong affinity of platinum (Pt) for amine groups [26,27,28,29], we further explored the deposition of Pt onto branched Au NPs, resulting in the formation of AuPt bimetallic NPs with adjustable optical characteristics (Appendix A).

In our prior research, we successfully synthesized branched AuPd NPs with high efficiency, with tunable Pd shell thicknesses of up to 2 nm. However, to maintain their long-term stability, additional amounts of PAH 50 K were required for post-synthesis stabilization.

Taking into account previous findings that suggest that for the same mass of polymer, low molecular weight ones may improve the stabilization of metallic NPs due to a higher number of single chains [30,31], we performed the synthesis of branched Au NPs using PAH with a lower molecular weight (17.5 K) as a shape-inducing agent. Branched NPs with a customizable optical response can be manufactured from PAH with a shorter chain length simply by adjusting the reaction conditions, such as pH, seeds, and polymer concentration. For instance, we obtained NPs with an LSPR band tunable from approx. 500 nm to 1000 nm (Figure 1a) by changing the concentration of seeds using PAH at a pH of 5.7. High-resolution transmission electron microscopy (HRTEM) images showed a similar branched morphology when compared with NPs produced with PAH 50 k (Figure 1b) [10]. The NPs showed long branches with rounded tips (Figure 1c,e), with tips presenting polycrystalline characteristics and multiple twinning planes (Figure 1d,f).

Using branched Au NPs with an LSPR response centered at approx. 1000 nm as a template, we investigated the deposition of Pt to create bimetallic AuPt branched NPs. After Au NPs synthesis, potassium tetrachloroplatinate(II) (K_2_PtCl_4_) and ascorbic acid (AA) were subsequently added to a suspension of as-synthesized, branched Au NPs in the same reaction vessel.

We investigated three different Au:Pt ratios (1:0.1, 1:0.5, and 1:1). Following synthesis, the NPs exhibited plasmon damping consistent with the Pt concentration. Those with an Au:Pt ratio of 1:0.1 displayed damping without significant displacement of the main LSPR band. As Pt content increased, damping became more pronounced, accompanied by a red-shift as a consequence of the increased Pt amount on the Au nanostructure, as previously observed [23] (Figure 2a).

The oxidation state of the metals in AuPt (1:1) NPs was studied through X-ray photoelectron spectroscopy (XPS). The survey spectrum confirmed the presence of gold and platinum through the respective binding energies (Appendix A). The spectra presented the typical doublets for the Au 4f XPS at 83.75 and 87.5 eV, indicating the presence of Au (0) [32] (Appendix A). The signals in the Pt 4f region displayed two peaks at 70.8 and 74 eV, attributed to Pt (0), two other peaks at 72.2 and 75.4 eV, assigned to Pt (II) [33,34], and finally, two peaks at 74.6 and 77.1 eV, assigned to Pt (IV) [35,36] (Figure 2b). The Au:Pt 1:1 NPs surface ratio obtained through XPS was ca. 1:0.8, in which 52.3% corresponded to Pt (0), 39.8% corresponded to Pt(II), and 7.9% corresponded to Pt (IV). The presence of oxidized or ionic Pt on the NP surface may contribute to the affinity of the nanozymes for oxygen, thus making them suitable candidates for catalytic reactions involving oxidation processes [37]. Through inductively coupled plasma (ICP) analysis, we obtained a total result of 89.9% of Au and 10.9% of Pt (Appendix A).

The morphology of AuPt NPs was studied through high-resolution transmission electron microscopy (HRTEM). No evident increase in size was observed in the AuPt NPs (Figure 3). The NPs exhibited a branched morphology, with an average size ranging between 120–150 nm (Figure 3a). Notably, while maintaining the branched structure, the surface of the branches appeared rougher and more granular (Figure 3b–d). Further analysis at high resolution revealed regions of branches showing Pt (111) on the surface, with lattice fringes showing an interplanar distance of 0.226 nm (Figure 3e,f). However, a homogeneous metallic Pt shell around the NPs was not always observed, as some regions displayed the characteristic crystalline features of Au without apparent Pt deposition (Appendix A). Our results seem to indicate a Pt deposition via heterogeneous nucleation and an island growth similar to previously reported Au@Pt nanospheres [38].

Some nanoparticles’ branches seemed to exhibit a shell, alongside metallic Pt, on the surface, which lacked typical metallic stacking planes (see Appendix A). Notably, this feature was absent in pure Au NPs. This observation suggests the possible formation of complexes between the amine-rich surface and ionic Pt [39].

The distribution of Pt along the NPs was analyzed through STEM energy-dispersive X-ray spectroscopy (EDX) analysis (Figure 4 and Appendix A). The branched NPs exhibited a heterogeneous distribution, with regions of higher Pt content dispersed across the entire nanostructure, revealing the intricate spatial arrangement of Pt. Our observation seems to align with previous calculations that identified a tendency of segregation, rather than mixing, in AuPt bimetallic NPs [40,41] and with other reports of spherical core-shell AuPt NPs [42].

To gain a deeper understanding of the catalytic capabilities of AuPt NPs and to generate interest in their promising applications, we investigated the modification of the surface chemistry, introducing carboxylic acid functionalities utilizing poly(acrylic acid-co-maleic acid) (PAcMA). AuPt NPs, which present a positive ζ-potential (+37.5 eV), underwent a surface charge change (ζ-potential −36 eV) after functionalization, demonstrating the presence of PacMA [10] (Figure S6).

### 3.2. Catalysis Results

The catalytic capacities of natural peroxidases and their synthetic counterparts depend on their remarkable ability to perform the oxidations of various chemical compounds using H_2_O_2_ as a co-substrate. This catalysis involves the cleavage of H_2_O_2_, generating highly reactive intermediates for substrate oxidation. Notably, studies have shown that both Au and Pt NPs can mimic the peroxidase-like activity of enzymes, and their combination may result in a highly efficient oxidase mimetic nanomaterial [43,44]. With this concept in mind, the catalytic efficacies of Au and AuPt branched NPs were evaluated using the oxidation of colorless TMB to its blue counterpart, ox-TMB, in the presence of hydrogen peroxide as a model reaction under ambient conditions. Au and AuPt branched NPs exhibiting a primarily localized surface plasmon resonance (LSPR) at approx. 1000 nm were selected for investigation. Additionally, after PAcMA functionalization, the Au and AuPt NPs were employed to discern differences caused by the surface potential in the catalytic results. The TMB oxidation mediated by the nanoparticles was analyzed using UV-Vis absorption spectroscopy to evaluate the nanomaterials’ peroxidase-like activity. Experiments were carried out across a range of TMB and H_2_O_2_ concentrations, utilizing 10 μg of Au enzyme and 5 μg of AuPt (refer to materials and methods). Notably, upon employing 10 μg of AuPt, the color transition occurred within seconds, even at lower TMB concentrations, hindering the ability to effectively compare data from both nanozymes. Subsequently, the initial reaction rates (V_0_) were determined by identifying the initial linear segment in different curves, utilizing a molar absorption coefficient of 39,000 M^−1^ cm for ox-TMB. V_0_ was plotted against the TMB concentration for the different nanomaterials to comprehensively characterize the nanoparticles as catalysts. Finally, the Lineweaver–Burk equation was employed to estimate the constants (K_m_) and maximum reaction velocity (V_max_). The obtained steady-state kinetic parameters, K_m_ and V_max_, are listed in Table 1.

Au NPs showed a K_m_ and a V_max_ of 0.21 mM and 1.80·10^−8^ M·s^−1^, respectively (Appendix A). Conversely, when AuPt NPs were employed, the K_m_ and V_max_ were 0.42 mM and 73.2 M·s^−1^ (Appendix A). Despite a 2-fold increase of in TMB affinity, the reaction speed was improved to ca. 40-fold, compared with the Au counterpart. Similar results were obtained from AuPt nanoclusters in comparison with the Au NPs in the oxidase-like reaction with TMB as the substrate, in which the Pt showed a higher affinity but with exceptional catalytic results when compared with pure Au or Pt nanomaterials [43]. The result for hydrogen peroxide showed a low affinity for Au or AuPt NPs, but the V_max_ showed a 41-fold increase in the V_max_, compared to the Au counterpart. The affinities of Au and AuPt align with other Au or Pt peroxidase-like nano-enzymes in the literature [11,45]. Due to the low affinity obtained for the Au branched nanoparticles for H_2_O_2_ and similar to previous reports [45], it can be deduced that during the catalytic process, the TMB molecules may adhere to the surfaces of the NPs, where they contribute with their lone-pair electrons from their amine groups to the nanomaterials. This donation of electrons increases the electron density and mobility within the different nanoparticles, consequently expediting the transfer of electrons from the nanomaterials to H_2_O_2_. This enhanced electron transfer leads to the reduction of H_2_O_2_ to H_2_O under acidic pH conditions and accelerates the rate of TMB oxidation by H_2_O_2_. Additionally, the presence of Pt in the final structure makes the TMB oxidation process more favorable [45].

Although the K_m_ results may seem modest compared to natural enzymes (Appendix A), it is important to note that inorganic NPs have weaker interactions with substrates. In this case, with the simple addition of Pt, the reaction rate is highly improved, not only compared to the Au counterpart but also in comparison with natural enzymes or different enzymes present in literature (Appendix A). The presence of Pt, as expected, enhances their efficiency, despite the lower affinity with the substrates, and confirms the need to not only value the enzymes based on their K_m_ values [46].

Surface charge is an essential factor when we analyze the peroxidase-like activity of NPs. Different works have explored the peroxidase-like activity in positive or negatively charged nanomaterials [47]. The post-functionalization proposed here allows for a better comparison once the nanomaterials possess the same size, shape, and composition. Upon functionalization with PAcMA, both Au and AuPt NPs showed an increase in both the K_m_ and V_max_ (Appendix A and Appendix A), with the AuPt@PAcMA NPs being the most efficient catalyst for TMB oxidation. The negatively charged NPs may have shown a higher affinity towards positively charged TMB substrates. The affinity between the opposite charges of both nanozymes and substrate molecules may facilitates their charge transfer.

Additionally, and in line with previous research, the Pt nanoshell may facilitate the radical formation and subsequent acceleration in electron transfer [48]. In addition, the presence of a negative surface charge upon functionalization, even if it may contribute to shielding the NP surface, could increase the stability of the nanomaterials and lead to a lower TMB affinity but a higher reaction rate in both cases [47]. Regarding the affinity for hydrogen peroxide upon functionalization, the K_m_ values indicate an improved affinity in both cases, with the enhancement being more pronounced for the AuPt nanomaterials. Although this increase in affinity is accompanied by a slight reduction in V_max_ for AuPt, the catalytic activity of AuPt NPs remains significantly higher, being 10.8-fold greater than that of the Au NPs. This indicates their excellent catalytic performance. The AuPt NPs exhibit superior catalytic efficiency, compared to the HRP enzyme, showcasing their ability and versatility for use in peroxidase catalytic processes, despite the presence of either positive or negative charges on their surface.

## 4. Conclusions

This study demonstrated the successful synthesis of Au and AuPt branched NPs using small molecular weight PAH. Au NPs exhibited excellent tunability and served as templates for Pt deposition. Significantly, Pt deposition did not completely suppress the LSPR of the Au NPs. Instead, the LSPR intensity decreased proportionally to the amount of Pt added, resulting in plasmonically active bimetallic NPs. In tests mimicking HRP enzymatic activity via TMB oxidation, AuPt NPs displayed significantly enhanced catalytic performance, compared to pure Au NPs. Furthermore, the easy post/synthetic protocol to change the surface charge from positive to negative expanded the versatility of the nanomaterials to be tailored to different decided applications, showcasing their high versatility in future applications. This work expanded the synthetic protocols present in literature for plasmonically active nanoparticles. Future studies could explore how variations in Pt coating thickness influence the properties of these bimetallic nanocatalysts.

## Figures and Tables

**Figure 1 nanomaterials-14-01166-f001:**
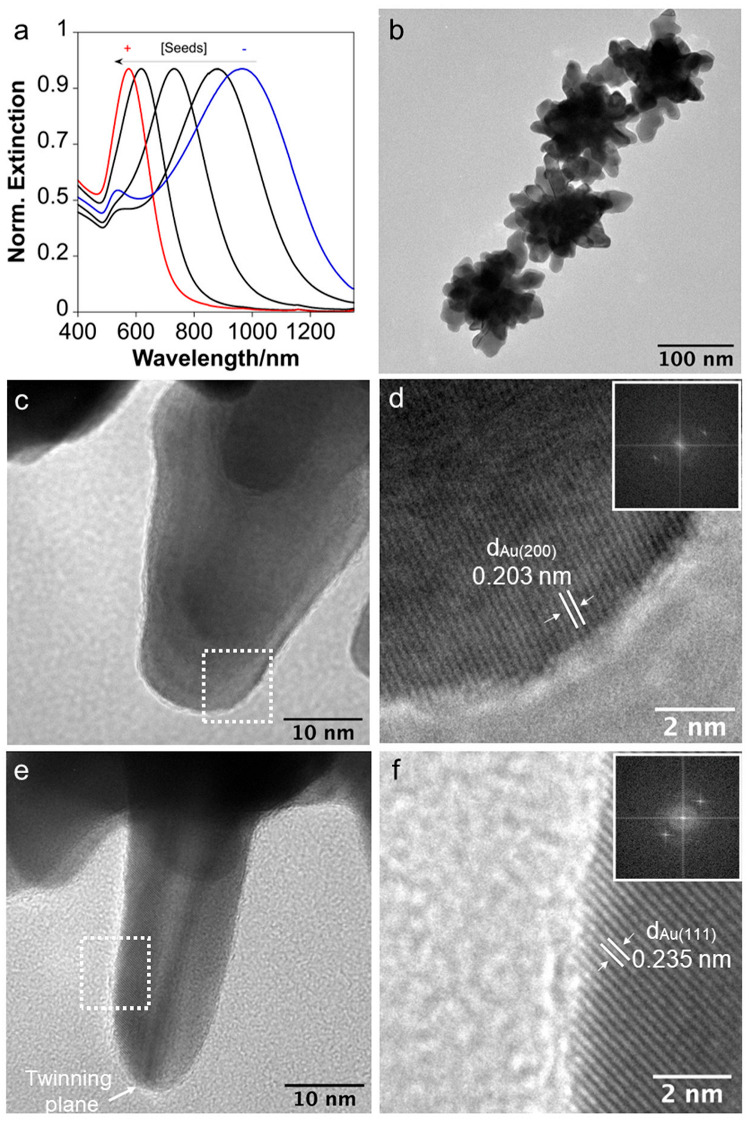
(**a**) Normalized extinction spectra of Au branched NPs obtained with different amounts of seeds ranging between 106.5 pM to 8.9 pM. (**b**) HRTEM of Au branched NPs obtained with 8.9 pM of seeds. (**c**) Details of a branch of an NP. (**d**) Details at a higher magnification and DDP of the NP details shown in panel (**c**). (**e**) Details of a branch of an NP. (**f**) Details at higher magnification and DDP of the NP details shown in panel (**e**).

**Figure 2 nanomaterials-14-01166-f002:**
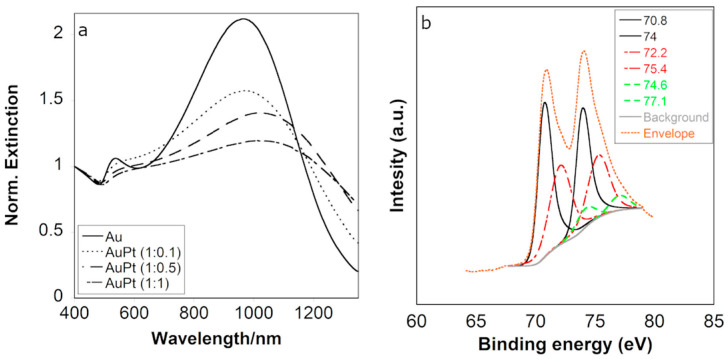
(**a**) Normalized extinction spectra of Au and AuPt NPs obtained with different Au:Pt molar ratios. (**b**) XPS spectra of AuPt NPs.

**Figure 3 nanomaterials-14-01166-f003:**
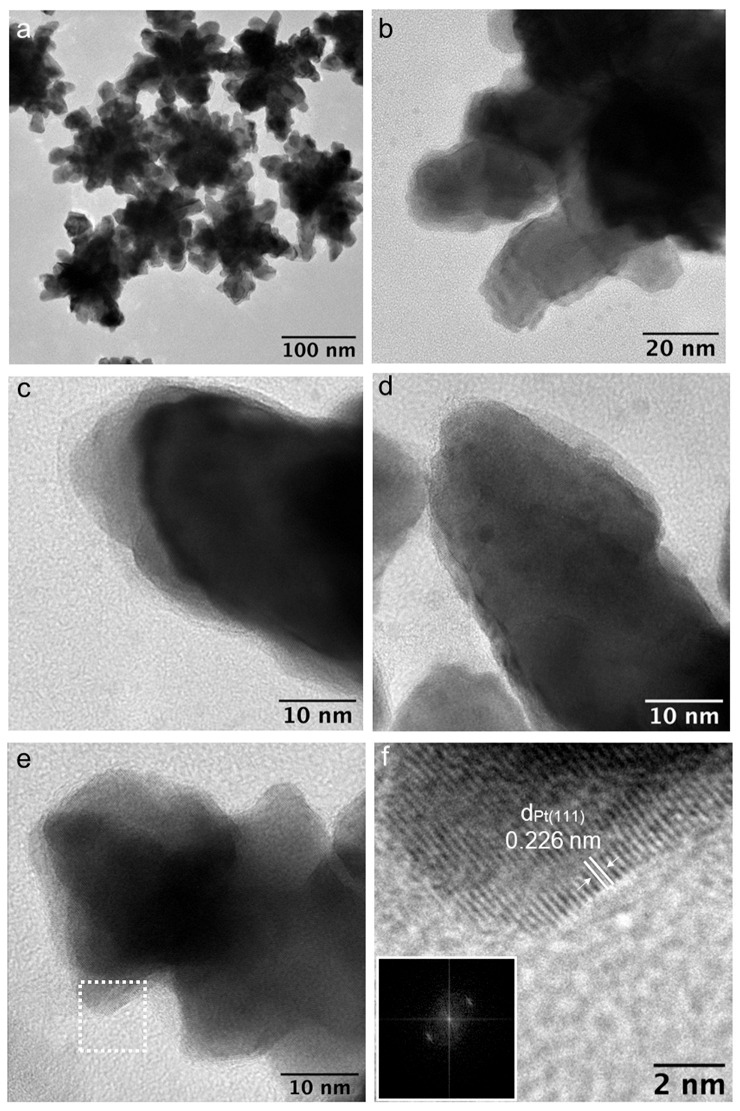
(**a**) HRTEM of branched AuPt NPs. (**b**) Details of a branched NP. (**c**–**e**) Details of branches of NPs. (**f**) Details of the area selected in panel (**e**) with interplanar distance and DDP.

**Figure 4 nanomaterials-14-01166-f004:**
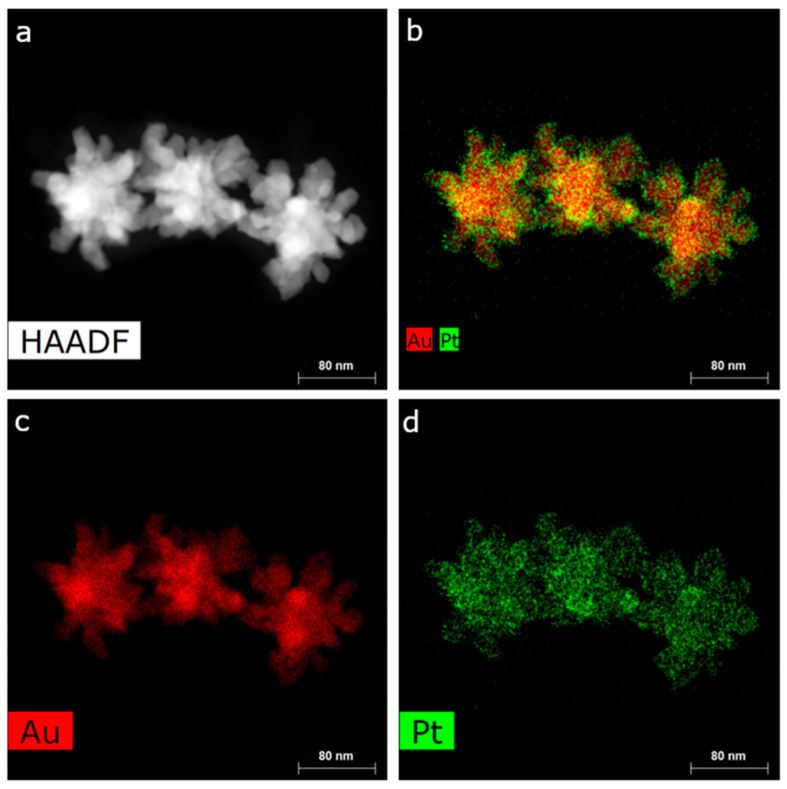
(**a**) HAADF-STEM image of AuPt NPs. (**b**) EDX map of AuPt NPs showing Au and Pt distribution. (**c**) EDX map of AuPt NPs showing Au distribution. (**d**) EDX map of AuPt NPs showing Pt distribution.

**Table 1 nanomaterials-14-01166-t001:** Summary of the apparent constant (K_m_), maximum reaction rate (V_max_) with TMB as the substrate, and H_2_O_2_ in acetate buffer (pH = 4.0) in the presence of the different bimetallic NPs.

	TMB		H_2_O_2_	
Nanoparticle	K_m_ (mM)	V_max_ (10^−8^ M·s^−1^)	K_m_ (mM)	V_max_ (10^−8^ M·s^−1^)
Au NPs	0.21	1.80	3229	1.72
AuPt NPs	0.42	73.2	1100.6	71.7
Au@PACMA NPs	0.22	2.78	3141	5.66
AuPt@PACMA NPs	0.80	146.4	291.5	58.7

## Data Availability

Data are contained within the article and Appendix A.

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
