# Peer review of "Synthesis, Structural Analysis, and Peroxidase-Mimicking Activity of AuPt Branched Nanoparticles"

_nanomaterials, 2024, doi:10.3390/nano14131166_

Round 1
Reviewer 1 Report
Comments and Suggestions for Authors
The article submitted by Nuti et al. devoted to the really interesting topic – Preparation of Bimetallic platinum-gold branched nanostructures for catalytic applications. The authors present presents a novel approach for synthesizing branched gold-platinum nanoparticles, utilizing polyallylamine hydrochloride stabilized branched gold nanoparticles with localized surface plasmon resonance (LSPR) as a template for platinum deposition. The authors shown the potential of the fabricated Au-Pt systems for increasing of peroxidase-like activity in the oxidation of 3,3',5,5'-Tetramethylbenzidine (TMB). As I already mentioned the topic is really interesting and actual, but during the reading of the paper there are quite some major issues arisen which are presented further. Therefore, I would recommend the presented manuscript for publication in the Nanomaterials journal only after the following issues are taken into account in a revised version.
Comments to authors:
- From the Introduction part it is not clear how the authors choose namely gold and platinum as metals for development of new catalyst. There is lack of information concerning other systems usually used for processes investigated. What is advantage of PtAu? The same question for the discussion part – the comparison of catalytic performance with other systems/materials has to be included.
- There is the lack of information concerning the origins of catalytic activity increasing when Pt added to the gold. What is the structure/nature of final bimetallic system? How does the Pt interact with the gold. It is well known that the Pt and Au are immiscible metals.
- Concerning the XPS data:
First – the original spectra should be included in figs. 2 and S1.
Second – it is not clear how the oxidative state of Pt influences the catalytic activity. If it should be the metallic Pt, Pt2+ or Pt4+ in order to improve catalytic activity
Third – what is about the catalyst after reaction – how the surface would change?
- Did the authors checked the reproducibility of this preparation technique?
Author Response
Reviewer 1
Q1.- From the Introduction part it is not clear how the authors choose namely gold and platinum as metals for development of new catalyst. There is lack of information concerning other systems usually used for processes investigated. What is advantage of PtAu? The same question for the discussion part – the comparison of catalytic performance with other systems/materials has to be included.
There is the lack of information concerning the origins of catalytic activity increasing when Pt added to the gold. What is the structure/nature of final bimetallic system? How does the Pt interact with the gold. It is well known that the Pt and Au are immiscible metals.
Answers 1_2: questions 1 and 2 are replied together.
We thank the reviewer for the comments.
Pt is one of the most used catalytic metals, while Au is more appreciated for its plasmonic properties and stability and combining these two characteristics could promote the use of such nanomaterials in a broader variety of sectors where both plasmonic and catalytic properties are desirable (i.e.: photocatalysis, sensing). We have improved the introduction to address this issue.
In our synthesis Au and Pt are not mixed, Pt is added and reduced on the surface of the Au nanoparticle during the second stage of the nanomaterial preparation to facilitate its deposition on the Au surface. We have observed how Pt deposition occurs via heterogeneous nucleation and island growth. Consequently, Pt is not forming a uniform shell but is distributed in regions on the Au surface. This distribution can provide more active sites for catalytic reactions to happen, while not completely quenching Au’s characteristic plasmonic properties. We believe that this issue has been addresses in section 2.1.
Regarding the comparison, we have performed the Au and AuPt catalysis for comparison for both, positively and negatively charged nanomaterials. Additionally, other peroxidase systems were reported in table S2, referenced in section 2.2, in a comparison between the results of the literature and the results present on this article.
Q3.- Concerning the XPS data:
First – the original spectra should be included in figs. 2 and S1
Second – it is not clear how the oxidative state of Pt influences the catalytic activity. If it should be the metallic Pt, Pt2+ or Pt4+ in order to improve catalytic activity.
Third – what is about the catalyst after reaction – how the surface would change?
Answer 3.- We thank the reviewer for the comments.
We have prepared a new Figure S1a) XPS survey spectrum of AuPt (1:1). We added the complete survey for the sample in the supporting information as Figure S2A.
The oxygen peak, centred in ca. 531.4 eV can be due to the presence of platinum oxides (DOI: 10.1116/11.20150202, /10.1021/acsomega.0c05644) The presence of sodium may be due to NaOH added during the growth of the branched gold nanoparticle and the chlorine due to the gold and platinum salt. There is no presence of any contaminant in the final reaction, for any constituent does not present in the reaction medium.
Regarding the first comment:
We added the following explanation in the main text:
The survey spectrum confirms the presence of gold and platinum through the respective binding energies (Figure S1a).
Answer to Q2: In our opinion, to be sure which is the better platinum oxidation state for perform the reaction, it would be necessary to obtain the same nanoparticle with only each one of the oxidation states to compare. We can affirm through literature review that different oxidation states of platinum may give better catalytic properties to the system in different reactions, as it is stated in the lines 116 to 118.
This comparison would require a different work package, to be sure that we only have one metallic state in the top of the particle. In this sense, our system may present here a limitation, once we use chemical reduction on the top of the template and ensure only one oxidation state with the same size and structure in the particle may be challenging. We just wanted to highlight that the presence of oxidized platinum is not a drawback, it can be a feature to take profit depending on what is the application of the nanoparticles that the future reader may be interested in.
Answer to Q3: We agree that would be interesting to analyse but highly challenging. The small amount added of the nanocatalyst to perform the catalyst reaction will not be enough to perform an XPS analysis. Additionally, the small amount of nanocatalyst (100uL. 10 ug) for a total volume of 2500 uL, would be challenging to separate from the catalysis medium and be sure that we do not have contaminations from the reaction medium. This work may require state of the art facilities not present in our laboratories at the moment, and we believe that this study will not change the outcome of the finding presented through the article.
Q4.- Did the authors checked the reproducibility of this preparation technique?
Answer Q4.- We thank the reviewer for the comment. The reactions were performed and analysed by different technicians’ multiple times to ensure reproducibility. Moreover, we have previously presented the synthesis of branched Au and AuPd NPs assisted by high molecular weight PAH. We can affirm our synthetic protocol is very well reproducible.
Reviewer 2 Report
Comments and Suggestions for Authors
This manuscript presents a straightforward method for preparing bimetallic platinum-gold nanostructures and explores their enzyme mimetic potential. While the research content holds significance in the field, the manuscript suffers from inconsistencies in analysis and result organization. Consequently, I recommend its publication in Nanomaterials only after major revisions. Several improvements are necessary to address the issues and enhance the manuscript's quality.
C1. The current title lacks impact and fails to highlight the innovative aspects of the research. For instance, "branched nanostructures" is not a distinctive description, and "catalytic applications" is too broad when referring to "peroxidase mimicking nanozymes." Utilizing a more specific and conceptual term could be more effective.
C2. The abstract does not adequately emphasize the novelty of the research.
C3. Including a figure that outlines the fabrication methods would facilitate easier understanding.
C4. The introduction should more clearly define the problem or challenge addressed by this research.
C5. Are Figures 1b and 3a truly HRTEM images of the samples? I think, they are simple TEM/STEM images where Figures 1d, 1f, 3f etc. are HRTEM images! Did the authors analyze the morphology of samples prepared with different Au ratios (1:0.1, 1:0.5, and 1:1)? The morphology of the optimized sample should be compared with all tested conditions.
C6. XRD analysis is recommended to better understand the crystal phases of the bimetallic nanostructures compared to single metallic nanoparticles.
C7. Please add a wide-range survey spectrum of the sample to help identify constituent elements, possible impurities, and their electronic properties.
C8. Providing evidence of radical formation and identifying the types of radicals through EPR analysis is recommended. Additionally, zeta potential measurements are necessary to demonstrate the charge properties of the materials.
Comments on the Quality of English LanguageModerate English language editing is recommended!
Author Response
Reviewer 2
Q1 The current title lacks impact and fails to highlight the innovative aspects of the research. For instance, “branched nanostructures” is not a distinctive description, and “catalytic applications” is too broad when referring to “peroxidase mimicking nanozymes.” Utilizing a more specific and conceptual term could be more effective.
A1.- We thank the reviewer for the comment. A new title has been proposed as:
Synthesis, structural analysis and peroxidase-mimicking active-site of AuPt branched nanoparticles.
Q2.- The abstract does not adequately emphasize the novelty of the research.
A2.- We thank the reviewer for the comment, the abstract has been slightly modified. The major modifications are highlighted below.
“ Bimetallic nanomaterials have generated significant interest across diverse scientific disciplines due to their unique and tunable properties arising from the synergistic combination of two distinct metallic elements. This study presents a novel approach for synthesizing branched gold-platinum nanoparticles by utilizing poly(allylamine hydrochloride) (PAH)-stabilized branched gold nanoparticles, with a localized surface plasmon resonance (LSPR) response around 1000 nm, as a template for platinum deposition. This approach allows precise control over nanoparticle size, LSPR band, and branching degree at ambient temperature, without the need for high temperatures or organic solvents. The resulting AuPt branched nanoparticles not only demonstrate optical activity but also enhanced catalytic properties. To evaluate their catalytic potential, we compared the enzymatic capabilities of gold and gold-platinum nanoparticles by examining their peroxidase-like activity in the oxidation of 3,3',5,5'-Tetramethylbenzidine (TMB). Our findings reveal that the incorporation of platinum onto the gold surface substantially enhances catalytic efficiency, highlighting the potential of these bimetallic nanoparticles in catalytic applications.”
Q.- Including a figure that outlines the fabrication methods would facilitate easier understanding.
A3.- We thank the reviewer for the comment, a figure representing the reaction scheme has been added to the supporting information.
Q4.- The introduction should more clearly define the problem or challenge addressed by this research.
A4.- We thank the reviewer for the comment, the introduction has been improved.
Q5.- Are Figures 1b and 3a truly HRTEM images of the samples? I think, they are simple TEM/STEM images where Figures 1d, 1f, 3f etc. are HRTEM images! Did the authors analyze the morphology of samples prepared with different Au ratios (1:0.1, 1:0.5, and 1:1)? The morphology of the optimized sample should be compared with all tested conditions.
A5.- We thank the reviewer for the comment. All the images were acquired with the HR instrument and with the acquiring conditions presented in section 3.5. The images 1b and 3a that the reviewer is addressing were acquired with the lower magnification presented in section 3.5. Whereas the images in figure 4 are STEM-EDX, as stated in the figure caption, and were acquired with the instrumentation specified in section 3.5
The sample that was fully characterized is the AuPt sample used for the catalytic experiments. The AuPt NPs obtained with other ratios were studied with UV-Vis spectroscopy to demonstrate the damping of the LSPR band proportional with the added amount of Pt. As the UV-Vis spectra maintains its characteristic shape when compared to the Au NPs and the AuPt (1:1), only with a different degree of damping, we believe we can confidently assume the morphology is maintained even in the NPs obtained with ratio 1:0.5 and 1:0.1.
Q6.- XRD analysis is recommended to better understand the crystal phases of the bimetallic nanostructures compared to single metallic nanoparticles.
A6.- We thank the reviewer for the comment. An attempt to characterise the sample by XRD has been done, but the amount of dry sample obtained in our case was insufficient to perform the analysis as indicated by the XRD technician. While X-ray diffraction would provide information about the crystalline planes throughout the entire sample, we believe that incorporating this analysis would not significantly alter the explanation or outcome of our study. The presence of different crystalline planes observed in the high-resolution transmission electron microscopy (HRTEM) images supports the evidences of the study.
Q7.-Please add a wide-range survey spectrum of the sample to help identify constituent elements, possible impurities, and their electronic properties.
A7.-We thank the reviewer suggestion. We added the complete survey for the sample in the supporting information as Figure S1A.
The oxygen peak, centred in ca. 531.4 eV can be due to the presence of platinum oxides (DOI: 10.1116/11.20150202, /10.1021/acsomega.0c05644) The presence of sodium may be due to NaOH added during the growth of the branched gold nanoparticle and the chlorine due to the gold and platinum salt. There is no presence of any contaminant in the final reaction, for any constituent not present in the reaction medium.
We added the following explanation in the main text:
The survey spectrum confirms the presence of gold and platinum through the respective binding energies (Figure S1a).
Q8.-Providing evidence of radical formation and identifying the types of radicals through EPR analysis is recommended. Additionally, zeta potential measurements are necessary to demonstrate the charge properties of the materials.
A8.- We thank the reviewer for the recommendation. Our primary objective with this manuscript is to present a novel and straightforward synthetic protocol for the synthesis of AuPt branched nanoparticles and to highlight their potential in catalytic applications. EPR studies are not accessible at the moment in our working facilities. The study of radical formation may be expected in future research jobs, in where different catalytic mediums as well as nanoparticle configurations may help to understand better the properties of the nanomaterials in different catalysis. In line to this support this affirmation, a new reference has been added to the text, in where the formation of radical were detected in AuPt nanomaterials, but not in Au or Pt nanomaterials alone, for the oxidation of TMB with H2O2: DOI: 10.1007/s00604-018-2981-5. We theorise a similar process happens with our system.
Regarding the charge properties of the materials, the zeta potential measurements have been presented in the main text, specifically in section 2.1, and further detailed in the supporting information in figure S5, before and after functionalization with PACMA. We believe this result are in line with the positive and negative surface charge expectations, and in line with the work performed.
Reviewer 3 Report
Comments and Suggestions for Authors
This paper presents a novel approach for the synthesis of branched gold-platinum (AuPt) bimetallic nanoparticles using poly(allylamine hydrochloride) (PAH) stabilized branched gold nanoparticles as a template. Overall, the manuscript is well-written and the findings are of interest. The paper can be accepted after the authors address the following questions:
I'm interested in the branch structure formed for this Au-Pt bimetallic nanoparticle. For some bimetallic systems like Au-Ag, they usually form a core-shell structure, while for Au-Pd, both core-shell and branched structures have been reported. Can the authors explain why the Au-Pt system tends to form a branched structure in this case? What are the factors that govern the formation of the branched morphology over a core-shell structure?
How do the catalytic properties of the AuPt branched nanoparticles, as demonstrated by the peroxidase-like activity, compare to other bimetallic nanoparticle systems reported in the literature? Can the authors provide a more quantitative comparison of the catalytic performance, such as the enhancement factor or catalytic rate constants, to better contextualize the significance of the observed improvement in activity.
Author Response
Reviewer 3
Q1.- I'm interested in the branch structure formed for this Au-Pt bimetallic nanoparticle. For some bimetallic systems like Au-Ag, they usually form a core-shell structure, while for Au-Pd, both core-shell and branched structures have been reported. Can the authors explain why the Au-Pt system tends to form a branched structure in this case? What are the factors that govern the formation of the branched morphology over a core-shell structure?
A1.- We thank the reviewer for the comment. In the work we are presenting, the AuPt NPs exhibit a branched morphology, rather than a spherical core-shell morphology, due to the template-mediated synthetic protocol employed in our study. In our synthesis, Au branched nanoparticles are initially synthesized, which subsequently serve as a template for the deposition of Pt. As a result, the branched morphology of the final AuPt nanoparticles is directly derived from the initial Au template. Controlling the amount of added Pt, it is possible to obtain the deposition of Pt on top of the template without quenching completely the plasmonic band. We believe that this process is advantageous, once it allows us to pre-define the size and the branching of the particles in an accurate way.
Q2.- How do the catalytic properties of the AuPt branched nanoparticles, as demonstrated by the peroxidase-like activity, compare to other bimetallic nanoparticle systems reported in the literature? Can the authors provide a more quantitative comparison of the catalytic performance, such as the enhancement factor or catalytic rate constants, to better contextualize the significance of the observed improvement in activity.
A2.- We thank the reviewer for the comment. We have presented a comparison with other systems applied for the same model reaction in table S2 in the supporting information. We have improved the explanation through the main text.
Round 2
Reviewer 2 Report
Comments and Suggestions for Authors
Thank you for your prompt revisions and thorough responses to the comments provided. The manuscript has been significantly improved and now presents a clear and compelling study on the synthesis, structural analysis, and peroxidase-mimicking activity of AuPt branched nanoparticles.
Your detailed attention to the suggested modifications has enhanced the clarity and scientific rigor of the work. The revised figures and additional explanations have effectively addressed the previous concerns.
I am pleased to inform you that, after reviewing the revised manuscript, I find it suitable for publication. Please consider the following points for future work:
- There might be no need to use a full-stop (.) at the end of the title.
- The comment on HRTEM was not about the instrument type but the image-taken mode. For example, Figure 1f was taken in HR mode.
Congratulations on your excellent work.
Author Response
Dear Reviewer,
We appreciate your comments on our revised version. This second Revised R2 version modifies both issues. The final dot in the title has been removed, and the HRTEM has been added to the caption of Figure 3 and comments.